# Progress on Catastrophic Health Expenditure in China: Evidence from China Family Panel Studies (CFPS) 2010 to 2016

**DOI:** 10.3390/ijerph16234775

**Published:** 2019-11-28

**Authors:** Xiaochen Ma, Ziyue Wang, Xiaoyun Liu

**Affiliations:** 1China Center for Health Development Studies, Peking University, Beijing 100191, China; xma@hsc.pku.edu.cn (X.M.); ziyuewang@bjmu.edu.cn (Z.W.); 2Department of Health Policy and Management, School of Public Health, Peking University, Beijing 100191, China

**Keywords:** catastrophic health expenditure, inequity, health care expenditures, income, China

## Abstract

**Background:** To provide an updated estimate of the level and change in catastrophic health expenditure in China and examine the association between catastrophic health expenditure and family net income, we obtained data from four waves of the China Family Panel Studies conducted between 2010 and 2016. **Method:** We defined catastrophic health expenditure as out-of-pocket payments equaling or exceeding 40% of the household’s capacity to pay. The Poisson regression with robust variance and generalized estimated equation (Poisson-GEE) model was used to quantify the level and change of catastrophic health expenditure, as well as the association between catastrophic heath expenditure and family net income. **Result:** Overall, the incidence of catastrophic expenditure in China experienced a 0.70-fold change between 2010 (12.57%) and 2016 (8.94%). The incidence of catastrophic health expenditure (CHE) decreased more in the poorest income quintile than the richest income quintile (annual decrease of 1.17% vs. 0.24% in urban areas, *p* < 0.001; 1.64% vs. −0.02% in rural areas, *p* < 0.001). Every 100% increase in income was associated with a 14% relative-risk reduction in CHE (RR = 0.86, 95% CI: 0.85–0.88) after adjusting for demographics, health needs, and health utilization characteristics; this association was weaker in recent years. **Conclusion:** Our analysis found that China made progress to reduce catastrophic health expenditure, especially for poorer groups. Income growth is strongly associated with this change.

## 1. Introduction

Around the world, there is an increasing commitment to achieving universal health coverage (UHC) [1]. Recent empirical studies have shown that coverage of essential health services has improved substantially, especially in developing countries [2,3]. However, studies have also shown that improved access is in conjunction with financial hardship as a result of receiving the healthcare that they need [4,5].

In the context of China, the world’s largest developing economy, the progress in access to health services co-occurs with the challenges of financial protection [6]. This phenomenon is consistent with findings from other developing countries [2,3]. Using data up to the first decade of the 21st century, China was listed as one of the bottom countries regarding catastrophic health expenditure (CHE)—the official Sustainable Development Goals(SDG) indicator of financial protection [7]—while also being one of the upper–middle performers in health service coverage [4]. In 2009, China launched a health system reform to establish an accessible, equitable, affordable, and efficient health system to cover all people by 2020 [8]. Despite the well-documented literature on the improved access to health care after the health reform [9], the understanding of financial protection in the post-reform era is still limited [10,11]. The nationally aggregated data on CHE has not been updated since 2011. Another alternative data source is micro-level survey data, however, few studies have reported national representative estimates after 2010. New data is needed to update the understanding of the exact level and change in incidence of CHE after China’s health system reform in 2009. Specifically, what is the current level of national incidence of CHE? How has the incidence of CHE changed in the recent past? What are the differences in levels and changes across different subgroups, for example, by income and area type?

This study contributes to the broader literature on CHE by focusing on the association between family net income and the incidence of catastrophic health expenditure. Existing studies mostly applied a static analysis by pooling cross-sectional data from different countries, and usually generated mixed pictures regarding this relationship [12,13,14,15]. China’s rapid economic growth provides a unique background to assess the relationship between income growth and the reduction in the incidence of CHE. Is economic growth associated with a linear change on CHE? In other words, is rising income related to a higher or lower incidence of CHE, and is there a specific level of income (threshold) that indicates pairwise relationship changes below and above the threshold? Furthermore, is the picture of this relationship different across different subgroups of interest?

This study uses a nationally representative household survey to investigate the trends of catastrophic health expenditure (CHE) after China’s 2009 health system reform. The objectives of this study are as follows: (1) To provide an updated estimate of the level and change in CHE at the national average, stratified by subgroups, and (2) to characterize the association between income growth and CHE reduction at the national average, stratified by subgroups.

## 2. Methods

### 2.1. Data Sources, Sampling Method, and Collection

For the analysis throughout the paper, we used four waves of data from the China Family Panel Studies (CFPS 2010, 2012, 2014, and 2016) [16]. The CFPS is a nationally representative, biennial household survey that has been performed since 2010, organized by the Institute of Social Science Survey, Peking University [17] (sampling protocol is publicly available at www.isss.edu.cn/cfps/). The sample of CFPS covered twenty-five provinces (excluding Hong Kong, Macao, Taiwan, Xinjiang, Tibet, Qinghai, Inner Mongolia, Ningxia, and Hainan), representing 94.5% of the total population in Mainland China. The population pyramids of the baseline cohort (CFPS 2010) showed consistency with China’s Census in 2010 [16]. The Peking University Biomedical Ethics Review Committee provided ethical approval of the survey (Approval number: IRB00001052-14010). All respondents read a statement that explained the purpose of the study and gave consent to continue.

The CFPS baseline cohort included 14,960 households in 2010. No new households entered the panel in the following wave, and the successful-tracking rates were 85%, 89%, and 89% for the 2012, 2014, and 2016 follow-ups. After excluding observations of missing data regarding total expenditure and spending on food or health services, our analytic sample included an unbalanced panel of 14,331 households (or 51,880 observations). The population pyramids of the dataset from the analytic sample still showed consistency with the raw dataset (Appendix A). Trained investigators conducted face-to-face interviews aided by computer-assisted personal interviewing (CAPI) technology. Automatic data checking was performed in every questionnaire. Of the households with completed questionnaires (60%), 25% were revisited by telephone, 15% by recording, 15% by on-site re-interview, and 5% by videotape to check the accuracy of the data.

### 2.2. Variables

#### Catastrophic Health Expenditure

Our outcome variable was catastrophic health expenditure (CHE). CHE occurs when a household has to reduce its basic expenditure to cope with out-of-pocket health expenditures (OOP) [18]. However, there is no consensus on the measure of CHE [19,20]. One strand of literature defines household out-of-pocket health expenditures as catastrophic when they exceed a given percentage of household income or consumption [12,18]. This approach is straightforward for calculation, but is also subject to underestimation of the real CHE, especially among low-income groups. Poor households have to spend a disproportionally higher percentage of income on food; therefore, their real capacity to pay is lower than that measured by total income or consumption [21]. To address this potential bias, other studies relate health expenditures to income or consumption less a deduction for necessities instead of total income or consumption, although this approach imposes more requirements for data availability [19,21].

Taking advantage of the rich household data from CFPS, we were able to calculate CHE based on the latter approach. Following previous studies [20,22,23], we defined CHE as an out-of-pocket payment (OOP) for health care that exceeded 40% of a household’s capacity to pay. Household capacity to pay was defined as the total household expenditure minus subsistence spending based on food expenditure, and the OOP payment was an estimation of aggregate health costs paid by the household with a 12 month recall period from the household head (see the Appendix A). To provide a more complete perspective of catastrophic health expenditure, we also reported CHE calculations based on total household expenditure (Appendix A) and the OOP budget share (OOP as the proportion of total household expenditure, Appendix A).

### 2.3. Covariates

Following previous studies, we included the following covariates in the analysis [15,23,24,25]: Socio-economic status of household head (age, sex, education, employment, and insurance), household structure (e.g., household with no children or older people, household with children but no older people, household with older people but no children, household with both children and older people, household with older people only), family net income (defined as gross income minus all the costs of income, e.g., farming costs, depreciation, and taxes, from a household’s revenues), urban/rural location, and year of interview of the household. In addition, we collected the following information for all household members: Health needs (e.g., self-reported overall health status, presence of chronic disease), health service utilization (e.g., outpatient visits in previous two weeks, inpatient service utilization during the past year), and type of health insurance.

### 2.4. Statistical Analysis

We used the χ2 test to test the overall trends across survey waves of 2010, 2012, 2014, and 2016. The annually family net income in Chinese Yuan (CNY) was expressed in 2010 prices using the consumer price index (CPI) and converted into US$ using the average 2010 exchange rate (i.e., 1 US$ = 6.77 CNY). We also performed stratified analysis by income quintile and urban–rural residence (according to the definition of the National Bureau of Statistics of China [26]).

We used Poisson regression with robust variance and generalized estimated equation (Poisson-GEE) to quantify the risk ratio (RR) of CHE in 2016 compared with 2010 to present the overall change in CHE over the six years. GEE was used for the analysis of the repeated observation of households from longitudinal data, which tended to be correlated [27], and Poisson regression avoided the overstatement of risk ratio estimated by the odds ratio (OR) from logistic regression when the outcome was quite common (≥10%) [28]. Adjusted variables included sex, age, education, employment of the head of household, age composition, urban–rural location, income group of the household, health status, health service utilization, health insurance of all household members, and regional effects of the provinces. We also performed stratified analyses to test the differences of the RRs (2016 vs. 2010) between different subgroups (e.g., income quintile and urban vs. rural residence) by introducing an interaction term between subgroup variables and year to the model.

In addition, we used the Poisson-GEE model to quantify the association between the RR of CHE and every 100% increase in family net income. We added the natural logarithm of family net income (logIncome) into the model as an explanatory variable. Similar covariates were adjusted as in the previous model. We also performed stratified analyses, as mentioned above.

All analyses were conducted using Stata version 14.0 (StataCorp, College Station, TX, USA). We expressed dichotomous data as risk ratios (RR) with 95% confidence intervals (95% CI), and continuous data as mean differences (MD) with 95% CI. All regression analyses were weighted based on family-level national sampling weights [17].

### 2.5. Ethical Approval

The CFPS is a nationally representative, biennial household survey that has been performed since 2010, organized by the Institute of Social Science Survey, Peking University. The Peking University Biomedical Ethics Review Committee provided ethical approval for the survey (Approval number: IRB00001052-14010). All respondents read a statement that explained the purpose of the study and gave consent to continue.

### 2.6. Availability of Data and Materials

The datasets generated and/or analyzed during the current study are available from the CFPS dataset, http://www.isss.pku.edu.cn/cfps/en/index.htm.

## 3. Results

A total of 51,880 household-wave observations (48.87% urban and 51.13% rural) from 2010 to 2016 were included in the analyses. The mean age for the household head was 39.30 years (SD: 22.38 years) in 2010, and 35.02% of the household heads were female. The mean family net income was 42253.73 CNY (6241.32 US$) per year, and the median family net income was 30,000 CNY (4431.31 US$) per year.

### 3.1. The Trend in the Incidence of CHE from 2010 to 2016

Between 2010 and 2016, the mean family net income in China increased by 63.7% from 33069.51 CNY (4884.71 US$) in 2010 to 54134.96 CNY (7996.30 US$) in 2016 (price after adjustment for inflation). During the same period, the proportion of households experiencing catastrophic health expenditure showed a 0.70-fold change (or decreased 28.88% in relative terms) from 12.57% in 2010 to 8.94% in 2016 (*p* < 0.001) (Table 1). Alternative CHE measures based on total household expenditure also yielded a constantly decreasing pattern (Appendix A). In addition, a continuous measure of OOP budget share also showed a decreasing pattern, although the most notable decrease occurred in the year 2016 (Appendix A). For the rest of our analyses, we focused on the outcome of CHE, as defined in our Methods section.

The gap in the incidence of CHE between different income quintiles narrowed. (Figure 1). In urban areas, the gap between the lowest household income quintile and highest household income quintile reduced from 14.75% (19.82% vs. 5.07%) in 2010 to 9.23% (12.88% vs. 3.65%) in 2016. A similar pattern was observed in rural areas, from 23.28% (30.04% vs. 6.76%) in 2010 to 13.50% (20.40% vs. 6.90%) in 2016. There was a larger reduction in the incidence of CHE in the lowest household income quintile than the richer quintiles. In urban areas, the mean annual decrease in the rate of CHE from 2010 through 2016 was 1.17% (95% CI: 0.67% to 1.66%) in the poorest income quintile compared with only 0.24% (95% CI: −0.02% to 0.51%) in the highest income quintile (*p* < 0.001). In rural areas, this difference was even greater: 1.64% (95% CI: 1.05% to 2.22%) in the lowest quintile and −0.02% (95% CI: −0.37% to 0.35%) in the highest quintile (*p* < 0.001).

Regression models adjusting for socio-economic status, health needs, and health utilization characteristics revealed that households in 2016 experienced a 37% relative-risk reduction in CHE than households in 2010 (overall RR for CHE, 0.63; 95% CI: 0.59 to 0.68 in 2016 vs. 2010) (Table 2). Generally, the magnitude of reduction in the incidence of CHE between 2010 and 2016 was negatively associated with family net income. In the lowest household income quintile, the relative-risk reduction in the incidence of CHE between 2010 and 2016 was 46% (RR, 0.54; 95% CI: 0.48 to 0.61). That estimate decreased to 26% (RR, 0.74; 95% CI: 0.58 to 0.94) in the highest income quintile. However, we did not observe a differential effect between rural and urban areas.

### 3.2. Association between Income Growth and CHE Reduction

Figure 2 illustrates the incidence of CHE by household income percentile using pooled data from 2010 through 2016. Higher income was associated with a lower incidence of CHE throughout the income distribution. Throughout the income distribution, rural households had higher rates of CHE than urban households. This gap in the incidence of CHE between urban and rural, however, tended to be narrower at higher income levels. In the bottom 1% of the income distribution, the incidence of CHE in rural areas was 11.61% (95% CI: 4.80% to 18.44%) higher than that in urban areas; in the top 1% of the income distribution, the gap between the two groups was 3.16% (95% CI: 0.42% to 5.90%).

Table 3 provides a quantitative association between every 100% increase in family net income and the reduction in the incidence of CHE. Adjusted for characteristics listed in the previous model and time effects, the regression model revealed that every 100% increase in family net income was associated with a 14% decrease in CHE (RR = 0.86, 95% CI: 0.85 to 0.88). This association attenuated over time; every 100% increase in family net income was associated with a 19% decrease in CHE in 2010 and a 14% decrease in CHE in 2016 (*p* < 0.001). The analysis stratified by urban–rural residence did not show any significant difference.

## 4. Discussion

Since the 1980s, China has experienced rapid economic growth. However, due to over-reliance on out-of-pocket payments (OOPs) for healthcare financing, the incidence of catastrophic health expenditures in China rose to 14% in 2008 [29]. This made China one of the countries with the heaviest burden of OOP spending in the world [4,12]. This study is, to our best knowledge, the first nationally representative study on the timely estimates of catastrophic health expenditure (CHE) after China’s 2009 health system reform. Herein, we highlight two salient findings from our analysis.

First, the overall declining trend of CHE and greater reduction in CHE among poor households are encouraging findings. Our 2010 CHE estimate of 12.57% was consistent with previous studies conducted up to the first decade of the 21st century (around 12.2% in 2003 and 12.9% in 2011) [6,30]. Taking advantage of an ongoing national survey of CFPS, we were able to measure CHE after 2010 and found China’s CHE constantly decreased to 8.94% in 2016. The reduction in the incidence of CHE in China has offered the world an encouraging counterexample in the global picture of a slight upward trend over recent years [12]. The difference between our results and the existing study may be due to the different measures of CHE (household capacity to pay approach vs. total household expenditure approach). However, we repeated our analysis using the latter approach as a sensitivity analysis and found the same pattern. Although poor households still bear the highest burden of CHE currently, they have also experienced the most considerable reduction in CHE over the past few years. Although identification of the exact reasons behind this progress was beyond the scope of this paper, we believed it is due to a combination of China’s constant economic development and the government’s commitment to poverty alleviation and policy efforts toward universal health coverage in the recent decade’s health reform [9,31,32]. This finding was particularly relevant given China’s commitment to eradicate poverty by 2020 [33], as high health costs were once the leading factor pushing households into impoverishment [11,29].

Second, we found a general pattern that lower income was significantly associated with higher CHE incidence. Our results contributed to a continuously and heavily debated question around the relationship between income and financial protection [12,13,14,15,34]. Some well-documented studies in low- and middle-income countries found a different pattern, i.e., that lower income was associated with lower CHE, although with different explanations. For example, when poor households in India chose to forgo healthcare to avoid health expenditures that might drag them into financial hardship, lower CHE was more likely to be observed in lower income groups [35]. Alternatively, when essential health services were sufficiently covered by social insurance in Thailand, health services that required personal payments were more likely to become luxury goods—for example, private high-end health services—thereby causing higher incomes to be associated with a higher incidence of CHE [25,36,37,38]. In the context of China, neither situation applies. More than 95% of the Chinese population has been covered by basic social health insurance schemes since 2011, which improved health service access even among the poorest [39]. However, the reimbursement rate is not sufficiently high, leading to a substantial proportion of essential health services paid for out-of-pocket [29,40]. If poor households in China do not have to forgo healthcare services but the reimbursement rates of their insurance plans are not high enough, lower incomes are more likely to be associated with a higher incidence of CHE in China. The situation was also observed in other middle-income countries, such as Iran [41,42], Brazil [43], and Mexico [44].

Our study has several limitations: First, the CFPS survey started in the year of 2010 and was followed-up for three waves until 2016, therefore this present study did not intend to explore a longer trend for CHE or a before–after analysis of the policy impact of China’s health system reform in 2009 on CHE. Instead, we focused on the important post-reform era between 2010 and 2016. Second, we did not have the data regarding indirect medical costs (e.g., accommodation, transportation) or opportunity costs (e.g., loss from job-leave and productivity), leading to our results being subject to underestimation of the real cost of medical expensed in China. Finally, we would like to acknowledge here that the present discussion regarding the association between income and CHE should not be interpreted as causal. Rather, the association found in this study underscored the need for research to understand health products and health financing in different economic development stages.

Although the decline in the incidence of CHE was very encouraging, policymakers should be cautiously optimistic about these findings. First, the current incidence of CHE in China is still one of the highest in the world [12,14,15]. To reach universal health coverage, China has a long way to go. Second, with the rapid increase in China’s aging population and the incidence of chronic diseases, both population aging and the epidemiological transition will exert tremendous pressures on healthcare financing in the future. Third, China’s economic growth has changed from a high-speed growth of more than 10% to a medium-speed growth of 6%. It remains unclear whether the decreasing trend of CHE alongside economic growth will continue in the future. Nevertheless, our analysis of the improvement in financial protection demonstrated an early appraisal of China’s recent health system reform toward achieving universal health coverage.

## 5. Conclusions

In summary, China made progress regarding the reduction in catastrophic health expenditure during 2010–2016, especially for poorer groups. Income growth is strongly associated with this change. Lower income was significantly associated with a higher CHE incidence.

## Figures and Tables

**Figure 1 ijerph-16-04775-f001:**
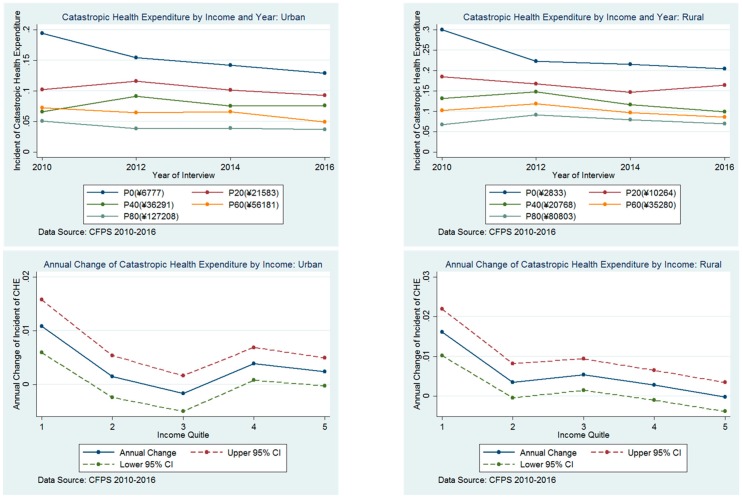
Level and change in incidence of catastrophic health expenditure by income group, 2010–2016.

**Figure 2 ijerph-16-04775-f002:**
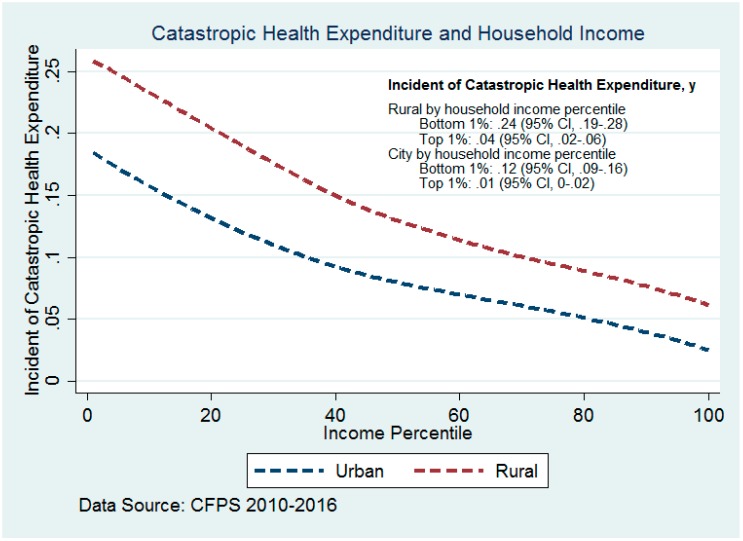
Incidence of catastrophic health expenditure by household income percentile ^a^, 2010–2016. a Household income is defined as “family net income”, which is the gross income minus all costs of income, e.g., farming costs, depreciation, and taxes from a household’s revenues.

**Table 1 ijerph-16-04775-t001:** Incidence of catastrophic health expenditure in China from 2010 to 2016.

Year of Interview	Mean Family Net Income [1]	Incidence of CHE (95% CI) [2]	Number of Households
2010	33,069.51	12.57% (11.61%, 13.53%)	12,851
2012	36,011.81	13.11% (12.08%, 14.15%)	11,477
2014	45,705.77	10.23% (9.45%, 11.01%)	13,668
2016	54,134.96	8.94% (8.29%, 9.60%)	13,884
Total	42,253.73	11.21% (10.71%, 11.71%)	51,880
Estimates of CHE based on previous references			
Meng et al. 2011 [6]	37,161.25	12.9% (NA)	18,800
Meng et al. 2008 [6]	24,654.47	14.0% (NA)	56,396
Li et al. 2008 [29]	NA	13.0% (NA)	55,556
Meng et al. 2003 [6]	16,566.98	12.2% (NA)	57,023

CHE: Catastrophic health expenditure. CI: Confidence interval. NA: Not available; Prices in 2010 CNY; *p* < 0.001 for linear trends of the incidence of CHE across years.

**Table 2 ijerph-16-04775-t002:** Risk of CHE in 2016 compared with 2010.

Subgroup	RR ^a^	*p* Value
**National Average**	0.63 (0.59–0.68)	NA
**Subgroup: Income Quintile**		
Poorest Quintile	0.54 (0.48–0.61)	Reference
Poor Quintile	0.66 (0.58–0.75)	0.024
Middle Quintile	0.73 (0.62–0.86)	0.003
Rich Quintile	0.62 (0.51–0.75)	0.208
Richest Quintile	0.74 (0.58–0.94)	0.021
**Subgroup: Urban/Rural**		
Rural	0.64 (0.59–0.70)	Reference
Urban	0.61 (0.54–0.68)	0.415

RR: Risk ratio. NA: Not applicable. ^a^ The regression models adjusted for adjusted for sex, age, education, employment of the head of household, age composition, urban–rural location, income group of the household, health status, health service utilization, health insurance of all household numbers, and regional effects of the provinces.

**Table 3 ijerph-16-04775-t003:** Association of CHE with every 100% increase in household income ^a^.

Subgroup	RR ^b^	*p* Value
**National Average**	0.86 (0.85–0.88)	NA
**Subgroup: Year of Interview**		
2010	0.81 (0.79–0.84)	Reference
2012	0.89 (0.87–0.92)	<0.001
2014	0.87 (0.85–0.90)	<0.001
2016	0.86 (0.84–0.89)	<0.001
**Subgroup: Urban/Rural**		
Rural	0.87 (0.85–0.89)	Reference
Urban	0.85 (0.83–0.87)	0.091

RR: Risk ratio. NA: Not applicable. ^a^ Household income is defined as “family net income”, which is the gross income minus all costs of income, e.g., farming costs, depreciation, and taxes from a household’s revenues; ^b^ The regression models adjusted for sex, age, education, employment of the head of household, age composition, urban–rural location, income group of the household, health status, health service utilization, health insurance of all household numbers, year of interview, and regional effects of the provinces.

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
