# Peer review of "Progress on Catastrophic Health Expenditure in China: Evidence from China Family Panel Studies (CFPS) 2010 to 2016"

_ijerph, 2019, doi:10.3390/ijerph16234775_

Round 1

Reviewer 1 Report

Dear Authors,

I really appreciate the extensive research you made. 

I just have one  remark:

In the Conclusions section, the description is too sparse. I would include three sentences from this part to the Discussion section and I'd call it Discussion/Conclusions. 

If you want to leave the Conclusions part, please write more. It is not very informative for the reader. For instance, refer to the introduction, and in just a few words summarise the achievements of your research.

Author Response

Thank you very much for pointing this out. We agree that the conclusion section is not very informative in our manuscript. And we appreciate that you offered two recommendation regarding the concluding messages. We accepted the first one that we included these three sentences to the discussion section. Please see revised text in Page 7, line 223 and Page 8, line 287 with the words highlighted in red.

Reviewer 2 Report

Dear Authors,

you made an incredible effort in presenting these results and doing such a study. Just two minors corrections:

- you could fill the Conclusions with some parts you have dedicated to the Discussion as the limitations and futher possible developments

- moreover have you detected similarities in another cases of study which can be compared to the Chinese example? However they are very well known the particularities of the Chinese case.

Good luck with the rest!

Author Response

- you could fill the Conclusions with some parts you have dedicated to the Discussion as the limitations and futher possible developments

Authors’ response: Thanks for this point! We actually got a similar feedback from another reviewer regarding the concluding message. The reviewer offered two recommendations: a) we either include three sentences from this part to the Discussion section and I'd call it Discussion/Conclusions, and b) write more if we need an independent conclusion section. Considering the word count, we decided to take the first recommendation that we included these three sentences to the discussion section change its name as Discussion & Conclusions. Please check Page 7, line 223 and Page 8, line 287 with the words highlighted in red.in red.

- moreover have you detected similarities in another cases of study which can be compared to the Chinese example? However they are very well known the particularities of the Chinese case.

Good luck with the rest!

Authors’ response: Thank you very much for this comment. We feel that we are on the same page regarding the interpretations of our results from China from a perspective of international comparisons. Actually, as we wrote in the discussion section, we did found many studies on CHE in low- and middle-income countries with similar or different patterns of the association between income and CHE. For example, in page 7 line 250 to 257 we compared the studies in India and Thailand:

Some well-documented studies in low- and middle-income countries found a different pattern that lower income was associated with lower CHE, although with different explanations. For example, when poor households in India choose to forgo health care to avoid health expenditure that might drag them into financial hardship, lower CHE are more likely to be observed in lower income groups [35]. Alternatively, when essential health services are sufficiently covered by social insurance in Thailand, health service that requires personal payment is more likely a luxury good—for example, private high-end health services—then a higher-income might be associated with a higher incidence of CHE [25, 36-38].

And we also provided examples in Iran, Brazil and Mexico between line 261 (page 7)  and 265 (page 8):

If poor households in China do not have to forgo the health care service but the reimbursement rate of their insurance are not high enough, lower income is more likely to be associated with a higher incidence of CHE in China.The situation is also observed in other middle-income countries as Iran [41, 42], Brazil [43], Mexico [44].

We believed our review on international literature is helpful for the readers to interpret our results in an international context, however, if the reviewer and/or editor believes we need more examples, we would like to do more literature to improve this interpretation. Again, thank you very much for this valuable point.
